# It Works! Organic-Waste-Assisted *Trichoderma* spp. Solid-State Fermentation on Agricultural Digestate

**DOI:** 10.3390/microorganisms10010164

**Published:** 2022-01-13

**Authors:** Carlotta Alias, Daniela Bulgari, Emanuela Gobbi

**Affiliations:** 1Agri-Food and Environmental Microbiology Platform (PiMiAA), Department of Molecular and Translational Medicine, University of Brescia, Viale Europa, 11, 25123 Brescia, Italy; carlotta.alias@unibs.it (C.A.); emanuela.gobbi@unibs.it (E.G.); 2B+LabNet-Environmental Sustainability Lab, University of Brescia, Via Branze 45, 25123 Brescia, Italy

**Keywords:** agro-industrial waste, fungal biomass, qPCR, circular economy, biofertilizer

## Abstract

This study aimed at valorizing digestate through *Trichoderma* spp. solid-state fermentation (SSF) to produce a potentially ameliorated fertilizer combined with fungal biomass as a value-added bioproduct. Plant-growth-promoting *Trichoderma atroviride* Ta13, *T. reesei* RUT-C30, *T. asperellum* R, and *T. harzianum* T-22 were tested on different SSF substrates: whole digestate (WD), digestate dried up with wood sawdust (SSF1), and digestate enriched with food waste and dried up with wood sawdust (SSF2). The fungal biomass was quantified by using a qPCR assay. The growth of the four *Trichoderma* spp. was only observed on the SSF2 substrate. The highest quantity of mycelium was produced by *T. reesei* RUT-30 (689.80 ± 80.53 mg/g substrate), followed by *T. atroviride* Ta13, and *T. asperellum* R (584.24 ± 13.36 and 444.79 ± 91.02 mg/g substrate). The germination of *Lepidium sativum* seeds was evaluated in order to assess the phytoxicity of the *Trichoderma*-enriched substrate. The treatments with 7.5% SSF2-R, 3.75% SSF2-T-22, and 1.8% SSF2-Ta13 equally enhanced the root elongation in comparison to the non-fermented SSF-2. This study demonstrated that digestate, mixed with agro-food waste, was able to support the cultivation of *Trichoderma* spp., paving the way to the valorization of fermented digestate as a proper biofertilizer.

## 1. Introduction

One hundred and eighty million tonnes of digestate are produced in the 28 countries of the European Union (EU-28) per year, as residue remaining after biogas production through anaerobic digestion. This process has been increasing steadily in recent years. Effectively, the number of biomethane plants in Europe has risen quickly from 187 plants in 2011 up to 725 plants in 2019 [1], and this trend is supposed to continue. More than 65% of this residue product is ascribable to agricultural digestate, typically a mix of manure and plants [2], whose current management is land application as fertilizer (Regulation (EU) 2019/1009) [3]. Interest in the application of digestate to agricultural soil has grown substantially given its good fertilizing value. Effectively, the key to soil fertility lies in the organic or humus content of soils. Agricultural digestate has a high content of organic matter in comparison to mineral fertilizer and could improve the health and structure of soil, besides also being rich in nutrients [4,5,6]. Furthermore, the use of digestate, reducing the need for mineral fertilization, could limit nitrate pollution to water, with a significant environmental advantage if properly managed [7]. Nevertheless, several potential drawbacks in digestate management still exist, namely the transportation issue (mainly linked to ammonia emissions during storage), the different land application techniques (ammonia emissions and nitrate leaching risks) [7], the highly variable composition [8], and the presence of high metal concentrations [9]. Moreover, the inappropriate in-field application of digestate can lead to the runoff of nutrients [9] and phytotoxic responses [10]. This last aspect is of high importance and closely related to the quality and quantity of the applied digestate [11,12].

It should also be mentioned that the EU Bioeconomy Strategy [13] emphasizes the need for the sustainable production of primary biomass and conversion of both primary and waste organic resources into food, feed, and bioenergy as well as other bio-based products. Although digestate has been widely studied as a biofertilizer (among others, [14]), few studies have been carried out on its valorization as a substrate of solid-state fermentation (SSF) processes for the production of other bio-products [15].

SSF is a biotechnological process that better matches the natural physiology of microorganisms than liquid fermentation, involving growth on solid substrates in the absence or near absence of free water, low to zero shear stress, and direct contact with gas phases [16]. SSF has arisen as a sustainable method for the production of large amounts of either fungal biomass or secondary metabolites, such as bioactive compounds and organic acids [17,18,19,20,21], in the concept of circular processes. Moreover, it is less energy-demanding for the sterilization and pre-treatment of solid agro-industrial wastes and requires less water and produces less wastewater with consequently lower environmental impacts [14].

In this study, the production of fungal biomass was addressed as it is a key point in the formulation and commercialization of microbial products, which have recently attracted considerable interest. As a matter of fact, the use of Plant-Growth-Promoting Fungi (PGPF) is a promising alternative to chemical fertilizer. Among the PGPFs, it is well recognized that *Trichoderma* spp. can effectively promote plant growth and root development [22,23]. Specifically, *Trichoderma* species display plant-growth-promoting versatility, as they can produce phytohormones, decompose organic matter, and protect plants from biotic and abiotic stresses [24]. *Trichoderma* spp. are used worldwide as biofertilizer fungi [24,25] to enhance crop growth and trigger plant systemic resistance to disease and tolerance to abiotic stresses under field conditions [26]. Moreover, *Trichoderma* are, among others, the most used fungi as a Biological Control Agent (BCA) against several pathogens [27]. 

In the industrial production of fungal biomass, processing steps, such as fungal harvesting, drying, formulation, and storage, are all tricky points able to reduce the number and the vitality of the microorganism [20,28]. As such, they can strongly influence the shelf life of microbial biopesticides and their bioefficacy in field [29,30]. Thus, there is a need for suitable organic substrates able to sustain an abundant fungal growth, be nutrient-rich, low-cost and readily available [31]. Agro-industrial wastes possess these features and can be used as substrates of SSF, gaining an economic value addition. Besides, specialty products such as agricultural inoculants might be better produced in SSF as it promotes higher spore production. Lastly, the enrichment of PGPF in an organic substrate (e.g., in the form of bio-organic fertilizer) before final application could facilitate its optimized performance and survival in the plant rhizosphere.

In the SSF approach, several parameters (e.g., inoculum and substrate composition) have been proposed for the production of different enzymatic compounds or secondary metabolites, but few efforts have been devoted to developing an easy method to describe and quantify fungal growth during fermentation. Fungal biomass in woody substrates was estimated by the measurement of ergosterol and chitin, components of the fungal cell membrane and cell wall, respectively. More recently, quantitative PCR (qPCR) using specific primers has also been used to detect DNA copy numbers of fungi in a target substrate as an index of abundance [32], which is especially useful in the SSF, where the fungal mycelium is inextricably entangled within the solid substrate.

This study aimed to ameliorate the agricultural features of the digestate through the cultivation of *Trichoderma asperellum* strain R, *Trichoderma atroviride* strain Ta13, *Trichoderma harzianum* strain T-22, and *Trichoderma reesei* strain RUT-C30 in SSF. The potentiality of the *Trichoderma*-enriched substrate as a new biofertilizer was assessed via (i) the set up of a genus-specific PCR method to quantify the fungal biomass entangled with SSF substrate and (ii) the *Lepidium sativum* toxicity assay to evaluate its effects on plants.

## 2. Materials and Methods

### 2.1. Digestate 

A sample of 10 L of agricultural digestate was collected at the end of the process at a full-scale biomethane plant operating in northern Italy. In Table 1, the input materials submitted to the 3-step anaerobic digestion process are summarized. The sample was kept at −20 °C until its use.

### 2.2. Solid-State Fermentation Substrates 

Three different digestate-based substrates were tested for the fungal solid-state fermentation (SSF): whole digestate (WD), digestate dried up with wood sawdust (SSF1), and digestate enriched with food waste and dried up with wood sawdust (SSF2). The different components were weighed and mixed in micropropagation containers (Microbox, Micropoli, Milano, Italy), as reported in Table 2. In detail, the whole digestate, including the solid and the liquid phase, was thawed and mixed thoroughly. Wood sawdust was a local carpentry by-product. Fruits no longer suitable for consumption were blended in a Waring blender until particles reached 1–2 mm in diameter. The substrates in microboxes were subjected to two consecutive cycles of sterilization (121 °C for 15 min) and allowed to cool down under a laminar flow hood for 16 h before fungal inoculation. 

### 2.3. Physical–Chemical Characterization of Substrates

The following parameters of WD and SSF2 before and after the fungal fermentation were determined: dry matter, moisture content and pH [33], porosity (measured as pore space), organic carbon [34], organic nitrogen [35], C/N ratio (calculated), ammonia nitrogen [36], and phosphorus and potassium [37,38].

### 2.4. Trichoderma spp. Culture Conditions

*Trichoderma asperellum* strain R, *Trichoderma atroviride* strain Ta13 (kindly provided by Prof. Z. Bouznad, Department of Botany ENSA, Algeria), *Trichoderma harzianum* strain T-22 (ATCC 20847), and *Trichoderma reesei* strain RUT-C30 (ATCC 56765) were routinely grown on slants of potato dextrose agar (PDA, Merck, Darmstadt, Germany) at 26 °C. For the SSF assays, 100 µL of conidia suspension (10^6^ in sterile distilled water) were seeded in each box and incubated at 26 °C and 60% RH under illumination of 12 h light/12 h dark cycles, using daylight tubes 24 W/m², 9000 lx in a climatic chamber (model 720, Binder, Tuttlingen, Germany) for 6 days. Similarly, non-fermented substrates were incubated as a control. The substrate colonization due to the fungal growth was monitored daily by visual inspection of the culture. Pictures were taken at the end of the incubation period.

### 2.5. Standard Curve for Mycelial Biomass Determination

To produce a standard curve for mycelial biomass versus the cycle threshold (Ct), *Trichoderma atroviride* Ta13 was grown in Potato Dextrose Broth (PDB) for 5 days at 26 °C under continuous shaking. The mycelium was collected, lyophilized, and ground to a fine powder. The DNA was extracted from 1–40 mg of dry mycelium and subjected to qPCR after a 1:100 dilution. qPCR was performed using the Real-Time PCR PowerUp SYBR Green Master Mix (Applied Biosystems, Waltham, MA, USA) in the 7500 Fast Real-Time PCR System (Thermo Fischer Scientific, Waltham, MA, USA). The oligonucleotide primers were designed to amplify a 130 bp fragment in the *Trichoderma* calmodulin (TCal) conserved sequence (forward: 5′-ACCGAAGAGCAGGTCTCTGA-3′ and reverse: 5′-CTCCTTGGTGGTGATCTGG-3′), as previously described [39]. To determine the PCR efficiency of the primer pair, a standard curve was generated using linear regression, and the Cq slope was calculated based on the Cq values for all dilutions (4 points, 10-fold dilutions from 0.1 to 100 ng) through the 7500 System software (v 1.4). Each PCR reaction was performed in three biological and technical replicates and a no-template control. 

### 2.6. Quantification of Trichoderma spp. Biomass

For quantification of mycelia mass, 5 g of fermented substrate was ground to a fine powder with a pestle after chilling in liquid nitrogen. Genomic DNA was subsequently extracted from 500 mg of powdered samples. Briefly, each sample was resuspended in 1 mL of extraction buffer (3% CTAB, 0.1% SDS), incubated at 65 °C for 10 min, and centrifuged at 8000 rpm for 5 min. An equal volume of chloroform/isoamyl alcohol (24:1) was added to the supernatant, mixed, and centrifuged at 6000 rpm for 5 min for two consecutive times. The DNA from the upper phase was precipitated by adding 0.6 volume cold isopropanol and incubating at −20 °C for 18 h. Samples were centrifuged at 12,000 rpm for 20 min and the pellets were washed with 500 µL 70% ethanol at −20 °C for 30 min. Samples were dried to remove residual ethanol, and the DNA was resuspended in 50 µL of TE buffer (10 mM Tris, 1 mM EDTA, pH 8). The extractions were performed in duplicate. The DNA was quantified spectrophotometrically and stored at −20 °C prior to Real-Time PCR amplification. qPCR with the TCal primer pair was conducted as described above.

### 2.7. Lepidium sativum Seed Germination and Root Elongation Assay

The assay was performed according to the Italian Environmental Agency guidelines [40], with some modifications. Briefly, seeds of *Lepidium sativum*, not treated with fungicides, were preliminarily checked for vitality in distilled water in the dark at 25  ±  1 °C (germination rates >90%). Substrates WD, 6 days-*Trichoderma* spp.-fermented SSF2, and unfermented SSF2 were tested at six doses (100, 30, 15, 7.5, 3.75, and 1.8% *w/w*), obtained by mixing them with a standard topsoil (ST). ST alone was used as a negative control. Three replicates per treatment were arranged by setting 20 g of substrate in 9 cm diameter disposable Petri dishes, covered with Whatman no. 1 filter paper, and wetted with distilled water. Ten seeds for each replicate were distributed on the filter. The three dishes of each replicate were packed into a tightly closed plastic bag and incubated at 25  ±  1 °C in the dark for 72 h. At the end of the incubation time, complete sprouts (≥1 mm) and root lengths were evaluated. The results were expressed as the mean root lengths ± standard deviation (SD). The statistical correlation between groups was analyzed via Student’s *t* test.

## 3. Results and Discussion

### 3.1. Evaluation of Fungal Growth on Different SSF Substrates

Solid-state fermentation is a promising tool for the production of microorganism biomass, fungal biocontrol agents, and biostimulants. This technique offers several advantages to the bio-pesticides industries due to high fermentation productivity, low water and energy requirements, and the opportunity to use waste as a substrate [16,19,41]. 

In a previous study on digestate as an SSF substrate, the mechanically separated solid phase was used as the sole nutrient source for the growth of edible fungi and enzyme production [42]. The digestate undergoes expensive and high energy consumption processes to separate solid and liquid phases [40,41]. 

The here presented SSF protocol allows the separate management of the two phases to be overcome, valorizing the whole digestate and reducing the energy input. The whole crude digestate was tested alone or in combination with other agro-industrial waste as substrate for SSF using four agriculturally relevant *Trichoderma* species. *Trichoderma*
*asperellum*, *T. atroviridae*, *T. harzianum*, and *T. reseei* strains did not show any growth on the whole digestate (Figure 1, substrate WD) as well as on the sole digestate added with the wood sawdust (Figure 1, substrate SSF1). These two substrates were unable to support fungal proliferation. The poor load of nutrients, such as sugars and organic acids, could be one of the reasons for this; in fact, the nutrient concentration in digestate is strongly influenced by the origin of the substrate and the management of the digestion process [9]. Moreover, as previously reported by Mejias and colleagues [15], other digestate characteristics (e.g., alkaline pH) are not conducive to the growth of *Trichoderma* spp. Conversely, all the tested *Trichoderma* spp. grew on the whole crude digestate mixed with agro-food waste and the wood sawdust (Figure 1, substrate SSF2). 

The added agro-food wastes were chosen because of their high sugar content that, in fact, sustained the fungal growth. Moreover, the addition of wood sawdust reduces the substrate humidity and increases the porosity (Table 3). In SSF, the solid substrate should have appropriate water content to support microbial growth and activity [43]. However, adding too much water might compact solid-substrate, impede oxygen transfer, and favor contamination. On the other hand, too little moisture would inhibit microbial growth and enzyme production and limit nutritional transfer [44]. 

To the best of our knowledge, this is the first report of fungal growth on whole crude digestate mixed with agro-industrial wastes. Some differences were observed in the growth behavior of the tested strains. *T. atroviride* Ta13 was able to colonize the substrate SSF2 in depth and produce conidia (Figure 2A,B), while *T. asperellum* R and *T. harzianum* T-22 proliferated and sporulated preferentially on the surface of the substrate. *T. reesei* RUT-C30 mycelium appeared as a weak structure, poorly penetrating the substrate (Figure 2C,D). The micromorphology and the different conidia productivity of the tested strains could explain these macroscopic differences [45,46]. 

### 3.2. Quantification of Trichoderma spp. Biomass Production in SSF

Biomass estimation is a key aspect of the evaluation of substrate suitability for the waste-based SSF sustainable production of BCAs or biofertilizer. Owing to the irreversible fungi–substrate binding that occurs in SSF, an indirect quantification method is needed [47]. Among several tools, qPCR offers the advantage to specifically detect and quantify fungal biomass in a complex matrix. The standard curve generated from the DNA concentration standards showed linearity between 1 × 10^−1^ and 1 × 10^2^ ng of DNA and Ct (Figure 3A). Ct values were plotted against log-transformed DNA amounts. The linear regression equation of the standard curve was y = −3.6088x + 20.8735, and the linear regression coefficient (R^2^) was 0.997. The PCR efficiency was 90%. The standard curve generated for the correlation between mycelial weights and estimated DNA showed linearity between 1 and 40 mg of dry mycelium (Figure 3B). The correlation coefficient (R^2^) of the linear regression was 0.9868. These curves, based on the *T. atrovi*ride Ta13 as a model of the fungal genus, were useful for the estimation of the biomass values of all the *Trichoderma* spp. tested.

The growth of *Trichoderma* species on substrate SSF2 was determined after 6 days of culture. Despite the visual appearance, the highest quantity of biomass was produced by *T. reesei* RUT-C30 with 689.80 ± 80.53 mg per g of substrate. On the other hand, *T. harzianum* T-22 produced the lowest amount of biomass, only 174.26 ± 2.87 mg per g of substrate (Table 4). These differences are likely attributable to the different growth rates of the various *Trichoderma* species tested, as well as to the different culture media and environmental conditions [48,49]. Solid-state fermentation is one of the best ways to obtain biomass and fungal spores, not only with high yield but also with good quality (e.g., viability and germinability) using waste such as grass powder, wheat and rice bran, sugar beet pulp, and cow dung [30,50]. The fungal biomass obtained, and the presence of conidia, suggest the possibility to use digestate mixed with agro-industrial waste as a substrate to produce a novel biofertilizer. As previously demonstrated, a *Trichoderma*-enriched bio-organic fertilizer better supports the plant growth than the sole *Trichoderma* suspension [51]. Moreover, the efficacy of BCA commercial products and bio-fertilizer is strictly related to the capability of the fungal biomass to survive several processing steps, including harvesting, drying, formulation, storage, and delivery. Further studies will be carried out to ascertain the fungal surviving capabilities along with the soil application steps.

### 3.3. Effects of Substrates on Seed Germination and Root Elongation

The root elongation of *Lepidium sativum* was evaluated after seedlings on whole digestate were treated with unfermented SSF2 and *Trichoderma*-spp.-fermented SSF2 at different concentrations ranging from 100% to 2% (Figure 4). 

Digestate was reported to have a phytotoxic effect at high levels (>20%) [12], while at low concentrations (2–3%), it is able to stimulate seed germination [52]. As expected, all the substrates showed strong phytotoxicity at the highest doses (100% and 30%), with root elongation well below the negative control value (30.5 ± 3.2 mm). WD supported the highest *L. sativum* root growth (45.9 ± 11.5 mm) at the dose of 1.8%, comparable to those indicated as a “good practice” dosages in agronomical use [12]. The performance of SSF2-NF as fertilizer was globally weak. The maximum mean root lengths (37.4 ± 6.3 and 36.0 ± 10.0 mm), obtained by treating the seeds with the two lowest doses, were slightly above the negative control. On the other hand, among the fermented substrates, SSF2-*T. asperellum* R demonstrated the best performance. In fact, two doses (15% and 7.5%) significantly increased the elongation of *L. sativum* roots (32.9 ± 1.9 and 48.9 ± 3.9 mm, respectively) compared to the same doses of both WD (22.2 ± 3.4 and 38.6 ± 9.18 mm) and SSF2-NF (23.3 ± 2.90 and 30.3 ± 8.39). Interestingly, even if beneath the negative control, the root elongation sustained by the dose of 30% of both SSF2-*T. atroviride* Ta13 and SSF2-*T. harzianum* T-22 was increased in comparison to WD and SSF2-NF (28.5 ± 3.4 and 20.6 ± 4.0 mm, respectively). The *T. reesei* RUTC30-fermented substrate showed a general higher phytotoxicity, most likely due to the largest amount of fungal biomass per gram of substrate (Table 4) and to the presence of high levels of secreted cellulolytic enzymes [45,53] that could interfere with the germination and the root elongation of *L. sativum* seeds. The overall best performances as root length enhancers were demonstrated by 1.8% WD, 7.5% SSF2-R, 3.75% SSF2-T-22, and 1.8% SSF2-Ta13 with mean length values above 40 mm. The observed effects of the fermented substrate might be related to (i) the different quantities of the fungal biomass (Appendix A), (ii) the different pattern of secondary metabolites secreted by the applied *Trichoderma* spp. [54,55] or to (iii) their potential attitude to detoxify contaminants present in the digestate [56]. Further studies will be carried out to set up the optimal dosage for each *Trichoderma*-enriched substrate and to identify the secondary metabolites pattern.

This new product composed of fungal biomass entangled with organic waste and digestate, enriched by fermentation products, could be suitable for in-field treatments. Moreover, in-field experiments will be performed to evaluate the efficacy of this novel bio-fertilizer on plant growth promotion and plant protection.

## 4. Conclusions

This study suggests an innovative use of whole digestate mixed with agro-food waste as a valuable substrate for fungal growth. This finding opens up a new way to use digestate as a low-cost, readily available substrate to obtain a large amount of fungal biomass via SSF. Moreover, the SSF process could be effortlessly scaled up on the farm, contributing to the recycling of agro-wastes, the production of a new tool to regenerate soil and the natural system, and the minimization of external inputs. In addition, after SSF, the *Trichoderma*-enriched digestate is less phytotoxic and it is able to sustain and enhance the root elongation of *Lepidium sativum* in a dose- and species-related manner.

In summary, the *Trichoderma* spp. enrichment of a digestate-based substrate allows for the valorization of biomass waste into valuable materials as, at the end of the SSF process, it has contextually (i) a lower pH, (ii) a large amount of PGPF inoculum, and (iii) lower phytotoxicity.

The here presented waste-valorizing process could lead to a useful agronomic product for improving crop growth, resistance to disease, and tolerance to abiotic stresses, paving the way for the new circular application of digestate.

## Figures and Tables

**Figure 1 microorganisms-10-00164-f001:**
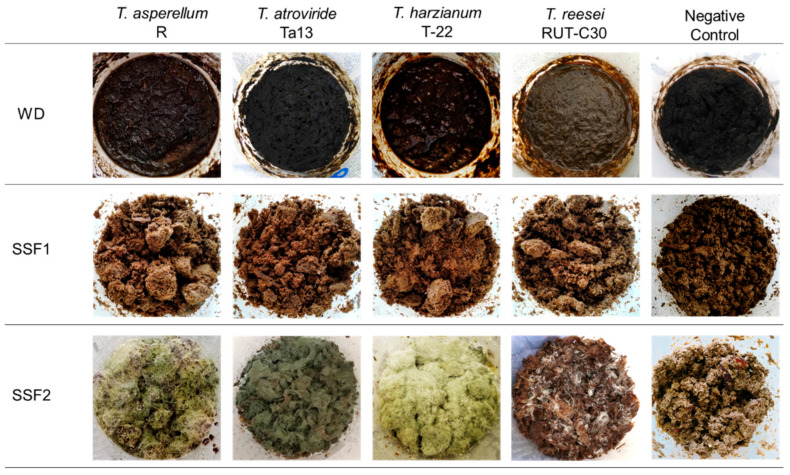
SSF of *Trichoderma* spp. after 6 days of culture on different substrates.

**Figure 2 microorganisms-10-00164-f002:**
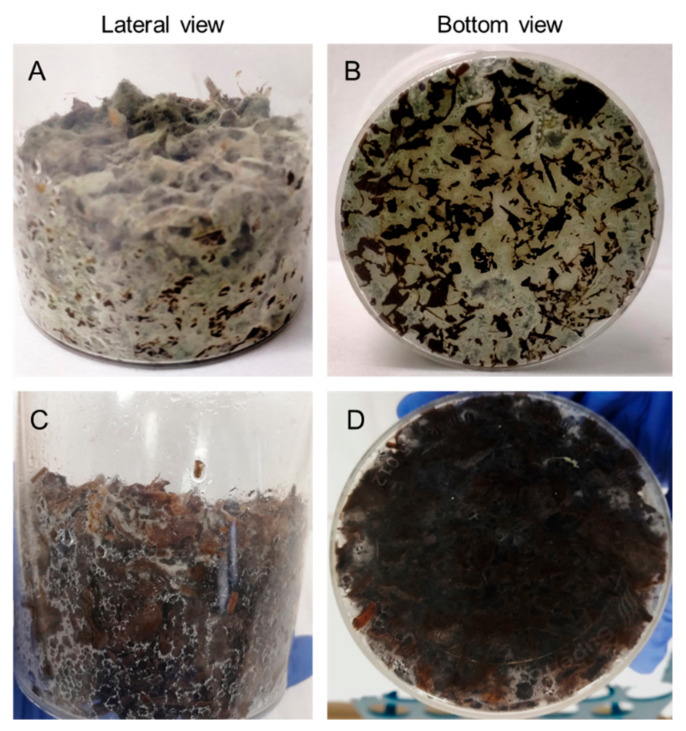
Details of *T. atroviride* Ta13 (**A**,**B**) and *T. reesei* RUT-C30 (**C**,**D**) growth on substrate SSF2 after 6 days of culture.

**Figure 3 microorganisms-10-00164-f003:**
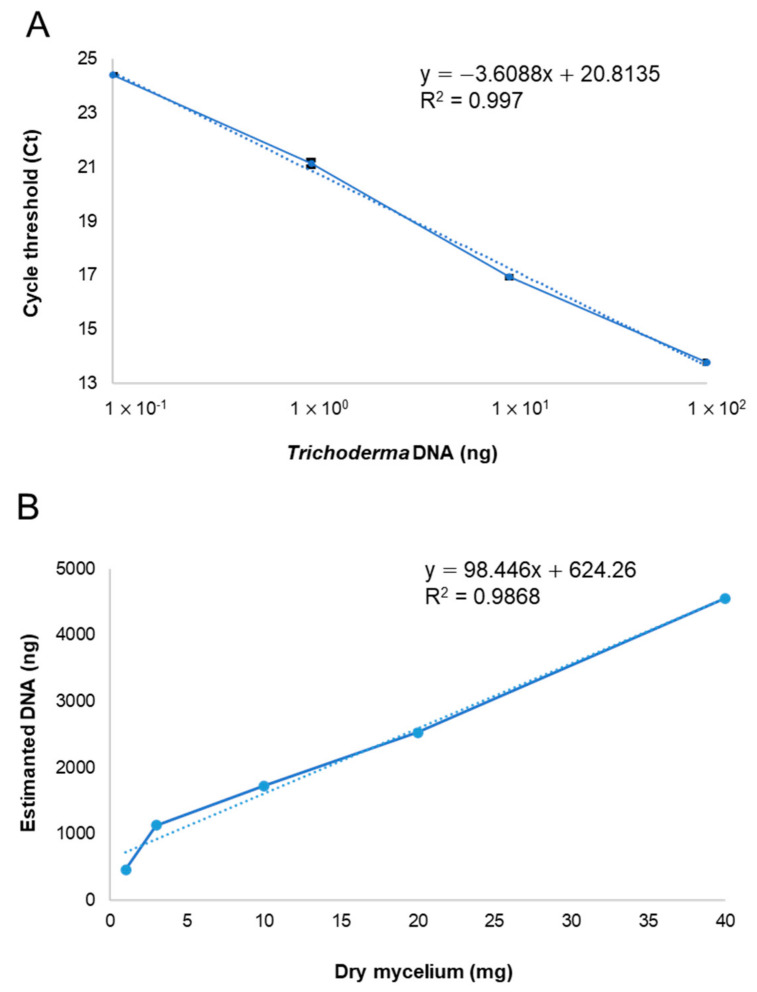
(**A**) Standard curve of *Trichoderma atroviride* Ta13 DNA concentration standards vs. cycle threshold (Ct) to quantify *Trichoderma* spp. using the TCal primer pair. (**B**) Linear relation between the *T. atroviride* Ta13 mycelial weights and the DNA estimation. Each PCR reaction was performed in three biological and technical replicates and a no-template control.

**Figure 4 microorganisms-10-00164-f004:**
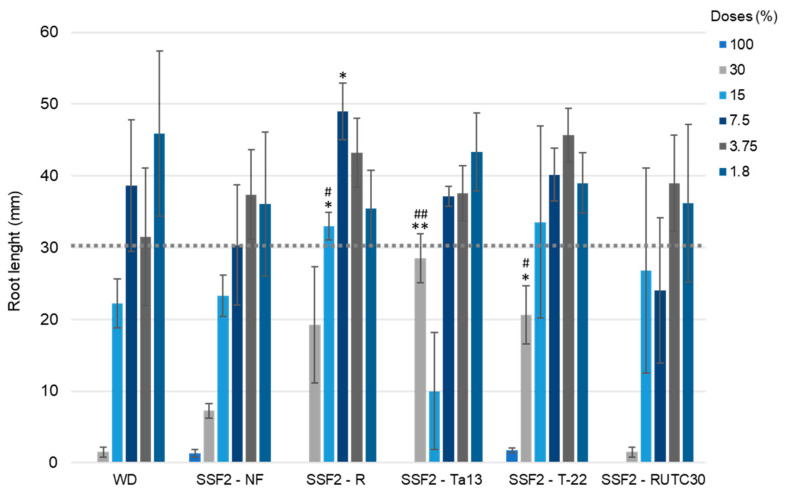
Root elongation of *Lepidium sativum* seeds treated with different doses of whole digestate (WD), SSF2 not fermented (SSF2-NF) and SSF2 fermented by *T. asperellum* R (SSF2-R), *T. atroviride* Ta13 (SSF2-Ta13), *T. harzianum* T-22 (SSF2-T-22), and *T. reesei* RUT-C30 (SSF2-RUTC30). Data are expressed as mean ± SD. Dotted line represents the root length of negative control (standard topsoil; mean value 30.5 ± 3.2 mm). Statistically significant versus WD according to Student’s *t* test: # *p* < 0.05; ## *p* < 0.01. Statistically significant versus SSF2-NF according to Student’s *t* test: * *p* < 0.05; ** *p* < 0.01.

**Table 1 microorganisms-10-00164-t001:** Biomethane plant feeding materials.

Constituents	% (*w*/*w*)
Dairy cattle slurry	38.1
Triticale	24.5
Silage corn stalks	13.9
Silage corn (1st harvest)	13.7
Silage corn (2nd harvest)	5.5
Fresh cattle manure	4.3

**Table 2 microorganisms-10-00164-t002:** Components of culture substrates for solid-state fermentation of *Trichoderma* spp.

Substrate	Constituents	% (*w*/*w*)
WD	Whole Digestate	100
SSF1	Whole Digestate	100
	Wood sawdust	~20 of the total weight
SSF2	Whole Digestate	70
	Apple—*Malus domestica*	10
	Banana—*Musa acuminata*	10
	Grape—*Vitis vinicola*	10
	Wood sawdust	~20 of the total weight

**Table 3 microorganisms-10-00164-t003:** Physico-chemical characterization of whole digestate (WD) and solid-state fermentation substrate not fermented (SSF2-NF) and fermented for 6 days (SSF2-F).

Parameter	Measure Unit	WD	SSF2-NF	SSF2-F
Dry matter (d.m.)	g/kg	81.3 ± 0.1	272.3 ± 0.1	169.4 ± 0.1
Moisture	%	91.9	72.8	83.0
Porosity	%	0.0	33.8	37.8
pH	unit	9.0 ± 0.1	8.5 ± 0.1	7.0 ± 0.1
Total organic carbon	% d.m.	41.0 ± 0.1	53.0 ± 0.1	49.1 ± 0.1
Total nitrogen	% d.m.	6.1 ± 0.1	3.3 ± 0.1	3.6 ± 0.1
C/N ratio	--	6.6	15.9	13.5
Ammonia nitrogen (NH^4+^)	% d.m.	3.13 ± 0.01	0.58 ± 0.01	0.46 ± 0.01
Phosphorus (P_2_O_5_)	% d.m.	0.06 ± 0.01	0.37 ± 0.01	0.36 ± 0.01
Potassium (K_2_O)	% d.m.	5.41 ± 0.01	1.45 ± 0.01	1.40 ± 0.01

**Table 4 microorganisms-10-00164-t004:** Biomass quantification of *Trichoderma* spp. expressed as mycelium per gram of substrate (mean value ± SD).

Fungi	Fungal Biomass (mg/g Substrate) ± SD
*T. asperellum* R	444.79 ± 91.02
*T. atroviride* Ta13	584.24 ± 13.36
*T. harzianum* T-22	174.26 ± 2.87
*T. reesei* RUT-C30	689.80 ± 80.53

## Data Availability

Not applicable.

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
