# Peer review of "It Works! Organic-Waste-Assisted Trichoderma spp. Solid-State Fermentation on Agricultural Digestate"

_microorganisms, 2022, doi:10.3390/microorganisms10010164_

Round 1

Reviewer 1 Report

Overall, this manuscript is relatively well prepared. Some suggestions or comments are provided for authors' consideration. 

  1. The Conclusion part can be improved to make it more attractive.
  2. Please provide the standard deviations for data in Table 1 and 2.
  3. Why we need to use Solid-State Fermentation (SSF)? Please provide some reasons for this point in Introduction section. 
  4. The key issue that the authors are trying to solve can be more clearly pointed out in Introduced. The current presentation is basically acceptable, but not very attractive.
  5. Please pay attention to the grammar mistakes and italic format of microbial names (e.g., Line 287-288). Please check the entire manuscript.
  6. In the manuscript title, the authors are suggested to specify the type of food waste and the type of digestate. That would be more instructive. 
  7. Can the authors provide a specific and whole name for Trichoderma spp.?

Author Response

Dear reviewer,

Reviewer 2 Report

Overall, the manuscript is good. However, I have some technical comments that need to be addressed before acceptance.

  1. Although the authors mention the objective of the research is valorization but it is not clear what specifically is valorization? Is it better root length as shown in figure 4 or the change in parameters shown in Table 3 or fungal growth? The data shown in figure 4 does not show a linear increase or decrease in root length indicating that some other unknown parameters might be involved.
  2. The difference in fungal biomass needs to be logically addressed. I recommend the authors test if the growth rate of T-22 strain is slower than others in the standard medium? Based on this, it appears that the phrase It works! in the title is too much extrapolation.
  3. The scientific names of the microorganisms should be italicized.
  4. Figure 3 forms the basis of quantification and it is not clear how many technical/biological replicates were used for analysis. It was not understandable how the primer efficiency was found to be 90%? Perhaps, it is a European format, but I would appreciate it if the authors can present the period as . not a comma in the figure. The same goes for figure 4.

Author Response

Dear reviewer,

Round 2

Reviewer 2 Report

The authors did a great job revising.